# Formulation and Evaluation of Mucoadhesive Oral Care Gel Containing *Kaempferia galanga* Extract

**DOI:** 10.3390/pharmaceutics16030421

**Published:** 2024-03-19

**Authors:** Indah Suasani Wahyuni, Irna Sufiawati, Amira Shafuria, Wipawee Nittayananta, Jutti Levita

**Affiliations:** 1Department of Oral Medicine, Faculty of Dentistry, Padjadjaran University, Bandung 40132, West Java, Indonesia; irna.sufiawati@fkg.unpad.ac.id; 2Oral Medicine Specialist Program, Padjadjaran University, Bandung 40132, West Java, Indonesia; mirashafuria@gmail.com; 3Faculty of Dentistry, Thammasat University, Pathum Thani 12120, Thailand; nwipawee@tu.ac.th; 4Department of Pharmacology and Clinical Pharmacy, Faculty of Pharmacy, Padjadjaran University, Sumedang 45363, West Java, Indonesia; jutti.levita@unpad.ac.id

**Keywords:** anti-oral mucosal ulcer gel, Carbopol 934, ethyl-*para*-methoxycinnamate, *Kaempferia galanga*, polyphenols

## Abstract

The rhizome of *Kaempferia galanga* (Zingiberaceae) is extensively used in traditional medicine by utilizing its various biological activities. It has been proven that ethyl-*para*-methoxycinnamate (EPMC) and other polyphenolic compounds are present in considerable amounts in the ethanolic extract of *K. galanga* rhizome (EKG). Our previous study confirmed that a dose of 0.5–1% of EKG demonstrated anti-inflammatory activity and a wound-healing effect in chemical-induced oral mucosal ulcers of Wistar rats. Currently, there are no reports on the formulation of oral gel containing EKG, thus revealing the potential of EKG to be developed as a herbal oral gel for mucosal ulcers. This study aims to formulate the best mucoadhesive oral care gel containing EKG in terms of physical stability. The presence of EPMC and the total phenols in the best EKG gel were also determined. The results revealed that Carbopol 934 is the best gelling agent for EKG gel preparations as proven by its stability during 14 days of storage. The statistical analysis resulted in a significant difference between the physical stability of the Carbopol 934-based EKG gel preparation compared to three commercial oral care gel products (*p* < 0.05). RP-HPLC chromatograms indicated that EPMC was identified in Carbopol 934-based gels containing 5% and 10% EKG at 6.056 and 6.146 min, respectively, with polyphenol levels of 1201.2557 mg/kg and 1849.1506 mg/kg, respectively. The hedonic test performed on 30 respondents to measure the degree of consumer acceptance and satisfaction confirmed that 5% EKG gel is the most sensorially accepted by the respondents. Data were analyzed using paired *t*-tests, one-way ANOVA, and a Kruskal–Wallis test. Taken together, the Carbopol 934-based gel containing 5% EKG could potentially be further developed as a topical anti-oral mucosal ulcer drug for clinical purposes.

## 1. Introduction

*Kaempferia galanga* L. (KG) (family Zingiberaceae), local name kencur, is a plant that originates from Asia. The rhizomes of KG have traditionally been used to lower blood pressure and relieve asthma, rheumatism, fever, coughs, headaches, stomach aches, and toothaches [1]. KG rhizome extract was reported to demonstrate pharmacological activities such as antioxidant, antimicrobial, anti-inflammatory, analgesic, amoebicidal, antinociceptive, anticancer, antituberculosis, antiangiogenic, insecticidal, and larvicidal effects [2,3,4].

Phytochemical studies show that KG contains various compounds such as terpenoids, phenolics, diarylheptanoids, flavonoids, polysaccharides, and essential oils [2]. Ethyl-*p*-methoxycinnamate (EPMC) is one of the bioactive substances contained in KG extract in large quantities. EPMC isolated from KG extract has activity against *Mycobacterium tuberculosis* and *Candida albicans* [5]. Umar et al. (2014) have confirmed the anti-inflammatory effects of EPMC on mouse and human macrophage cells. EPMC has been found to inhibit granuloma tissue formation in mice and block the release of inflammatory mediators such as interleukin-1 (IL-1) and tumor necrosis factor-α (TNF-α) in both animal models and cells [6]. EPMC undergoes hydrolysis into biologically active compounds in the human body, which in turn inhibits cyclooxygenase-2 (COX-2). COX-2 is the inducible isoform of the enzyme that catalyzes the conversion of arachidonic acid into prostaglandins, expressed in the presence of pathogenic stimuli. Prostaglandin biosynthesis increases in inflamed tissues and becomes a marker of acute inflammatory processes [7,8]. The inhibition of COX-2 by EPMC conduces an anti-inflammatory effect [5].

Preliminary research on the anti-inflammatory activity of ethanol extract of *K. galanga* L. (EKG) by Wahyuni et al. proved that EKG interacted with important amino acid residues in the catalytic site of COX-2 in an in-silico study and reduced the activity of the enzyme by in vitro method [4,9]. Secondary metabolites contained in the EKG, such as polyphenols, flavonoids, alkaloids, saponins, tannins, and triterpenoids, as well as EPMC play a role in that activity. These metabolites were detected using thin-layer chromatography (TLC), spectrophotometry, and high-performance liquid chromatography (HPLC) [10]. Further research has proven the anti-inflammatory activity and the potential of EKG to accelerate wound healing of chemically induced oral mucosal ulcerations in Wistar rats and was effective in low doses, between 0.5 and 1% [4]. In addition, other studies have proven the safety of *K. galanga* extract through in vivo studies of cytotoxicity. *K. galanga* extract and its main component can selectively exert an anticancer effect and show less toxicity toward normal cells [11,12]. Therefore, EKG may have the potential to be developed into a topical drug formula for oral mucosal ulceration therapy.

The main therapy for oral mucosal ulceration is corticosteroids [13]. Triamcinolone acetonide is the most frequently used corticosteroid commercially available in Indonesia in a gel formulation. However, the use of it has local side effects, such as oral candidiasis in 25–55% of patients, burning mouth syndrome, hypogeusia, oral hairy leukoplakia, and hypersensitivity reactions [14]. Recently, several non-steroidal anti-inflammatory topical drugs have become commercially available in Indonesia, and are indicated for oral mucosal inflammation, such as Gengigel^®^ containing 0.2% hyaluronic acid, Kincare^®^ containing 0.24% hyaluronic acid, and aloe vera extract (*Aloe vera*), as well as Alolair^®^ Plus, which also contains *Aloe vera* extract and sodium hyaluronate. These three commercial products have mucoadhesive properties that can increase the formation of a protective layer over the oral mucosal ulcer and moisturize, then enhance, wound healing.

Mucoadhesive polymers such as hydroxy propyl methyl cellulose (HPMC), hydroxy propyl ethyl cellulose (HPEC), Carbomer 934, sodium carboxy methyl cellulose (Na-CMC), benzocaine, tragacanth gum, and chitosan have been used in topical drug gel preparations for the oral mucosa. The addition of mucoadhesive polymers in drug formulations can prolong the contact time needed for the drug on the oral mucosa. The most effective combination of natural product and mucoadhesive was *Punica granatum* with Carbomer 934 polymer. This combination was effective in reducing the size of oral mucosal ulcers [15]. In terms of developing herbal medicines for anti-inflammatory oral mucosa, a combination with appropriate mucoadhesive materials is required. Then, it is necessary to determine the stability and the preferences for the drug formula that is being developed so that prospective and future predictions can determine whether the drug will be mass-produced and commercialized. Stability tests are usually carried out for 2 weeks to 1 month, to assess the physical properties of the drug and the stability of the formulation. Meanwhile, the hedonic test is a test in organoleptic sensory analysis that is used to determine the level of preference for a drug before distribution [16,17,18].

Research related to the formulation of oral care gel containing EKG has never been reported, thus providing potential novelty for this study. Moreover, by considering that *K. galanga* plants grow abundantly in Indonesia and have confirmed anti-inflammatory activity in animal models, our study aims to obtain the optimal gel formula of EKG, focusing on both physical properties and stability, as well as evaluating the preference of the gel by the panelists. The polyphenol content in EKG gel was also determined to ensure the presence of these active metabolites.

## 2. Materials and Methods

### 2.1. The Formulation of EKG Gel

This quasi-experimental technique was the synthesis of the most suitable gel formula for the EKG. The gel preparations consist of 3 formulas: F1, F2, and F3. F1 is composed of 5% EKG, Carbopol 934 (CAS number: 9003-01-4, Rahmat Putra CV, Bandung, Indonesia), glycerol (CAS number: 56-81-5, Fadillah CV, Bandung, Indonesia), propylene glycol (CAS number: 57-55-6, Quadrant Lab. CV, Bandung, Indonesia), triethanolamine (CAS number: 121-44-8, Brataco LLC, Bandung, Indonesia), and distilled water (CAS number: 7732-18-5, Brataco LLC, Bandung, Indonesia). In comparison, what was different in F2 was its EKG content of 10%, and in F3, it was its Na-CMC (9004-32-4, Brataco LLC, Bandung, Indonesia), which replaces the Carbopol.

Based on the results of our previous research, it was found that concentrations of 0.5% and 1% EKG showed the best activity in oral mucosal wound healing of rats; thus, the dose was equalized with humans [19] and resulted in doses of 5% EKG and 10% EKG for hedonic testing.

### 2.2. The Physical Stability Evaluation of the EKG Gel

The physical stability of EKG in Carbopol 934-based gel formulas (F1 and F2) and Na-CMC gel formula (F3) was evaluated by following previous methods [20,21,22], which comprised the organoleptic test, pH test, viscosity test, spreadability test, and homogeneity test. Organoleptic tests were carried out on day 0 and day 14. The homogeneity test was carried out by smearing 0.1 g of gel on a glass plate, while the viscosity test was carried out using the Sekonic Viscomate VM-10A-L sensor at 20 °C.

The spreadability test was carried out by placing 0.5 g of gel in a circular mark with a diameter of 1 cm on a glass plate. The second glass plate was placed on top of the first glass plate, then a 100 g weight was placed on top of both glass plates.

The physical stability evaluation was carried out at the Cosmetic Laboratory, Faculty of Pharmacy, and the Advanced Materials Laboratory, Department of Physics, Faculty of Mathematics and Natural Sciences, Padjadjaran University, from March to April 2023.

### 2.3. Materials for Determining Polyphenol and EPMC Identification in Gel Formula

Gallic acid (C7H6O5), Folin–Ciocâlteu (Sigma-Aldrich, Saint Louis, MO, USA) 10%, methanol pro analysis (Merck^®^, Darmstadt, Germany), sodium carbonate 7.5% (Na_2_CO_3_), aluminum foil, Magnetic Stirrer (IKA Yellow Line MAG HS 7), 100 mL beaker, watch glass, 10 mL, and 25 mL volumetric flask, 1000 µL micropipette, analytical balance, 1 mL and 5 mL volumetric pipettes, dropping pipette, test tube rack, 15 mL test tube with screw, water bath, and UV-visible spectrophotometry were used in determining the total polyphenol levels in gel F1 and F2 [23]. The Folin–Ciocâlteu reagent was used because this reagent reacts with the phenolic group in the plant extract to produce a blue color which can be measured at the absorption wave-length of 765 nm. The Folin–Ciocâlteu reagent oxidizes the polyphenol functional groups [24].

Ethanol absolute analytical grade (Merck^®^, Darmstadt, Germany), acetonitrile (Merck^®^, Darmstadt, Germany), methanol with HPLC grade (Merck^®^, Darmstadt, Germany), double-distilled water (API IPHA^®^, Jakarta, Indonesia), pure EPMC (Tokyo Chemical Industry Co., Ltd., Tokyo, Japan), PTFE membrane filters (Hawach Scientific^®^, Xi’an City, China), and Whatman™ filter papers (No. 1/120 mm and No. 41/90 mm) were used for HPLC analysis of the EKG gel formula. The reverse-phase HPLC procedure used an ODS column, water–acetonitrile (40:60) as the mobile phase, a flow rate of 1.0 mL/min, and detection at 308 nm. These procedures were carried out at the Central Laboratory of Padjadjaran University from March to April 2023.

### 2.4. The Hedonic Testing

The hedonic test, assessing the EKG gel for aroma, color, texture, and preference, was performed on the most stable gel. The Likert scale description of “very unlikely”, “unlikely”, “neutral”, “likely”, and “very likely” [18,25] was employed to obtain the score. Thirty respondents (21 females and 9 males) were recruited by purposive sampling from the population of the Faculty of Pharmacy students, at Padjadjaran University, aged 18–22 years. This test for humans has been approved by the Research Ethics Committee of Padjadjaran University (approval document number 898/UN6.KEP/EC/2023898/UN6.KEP/EC/2023).

## 3. Results

### 3.1. Physical Stability Evaluation of the EKG Gels

The physical stability evaluation of the EKG gels showed that the texture of all gels was thick and the odor matched that of the extract, which is associated with the EPMC content. The color of all gels was dark brown. However, after being stored for 14 days, only the F3 gel showed an unstable texture, and a change in consistency from gel to liquid (Table 1).

The homogeneity test revealed that the F1 and F2 gels fulfill the requirement, whereas the F3 gel is unstable, as indicated by bubbles on day 14 of observation (Table 2). Figure 1 shows the gel preparations on day 0 and day 14.

The pH test was carried out on all gel preparation formulations on day 0 and day 14. The pH value of all gel preparation formulations is presented in Table 3. F1, F2, and F3 showed a *p*-value > 0.05, or there was no significant difference in the pH level between all gel preparations between days 0 and 14.

The viscosity evaluation revealed that the comparison between F1, F2, and F3 resulted in a *p*-value > 0.05, thus confirming no significant difference in the viscosity of all gels between days 0 and 14 (Table 4).

The spreadability test showed that the comparison between F1, F2, and F3 resulted in a *p*-value > 0.05, thus confirming no significant difference in the spreadability for all gels between days 0 and 14 (Table 5).

F1 and F2 gel already proved their stability/homogeneity after being stored for 14 days. F1 and F2 were then compared in their physical properties with the commercial oral care gel preparations. Table 6 shows that F1 and F2 compared to the commercial oral care gels showed a *p*-value < 0.05, or there was a significant difference in the average pH between F1, F2, and all commercial oral care gels preparations.

The F1 and F2 gel preparations compared with commercial oral care gel preparations showed a *p*-value < 0.05 (Table 7), thus authenticating a significant difference in the average viscosity between F1 and F2 with commercial oral care gel preparations.

The F1 and F2 gel preparations compared with the commercial oral care gels showed a *p*-value < 0.05 (Table 8), thus authenticating a significant difference in the average spreadability between F1 and F2 gel preparations with commercial oral care gel preparations.

### 3.2. EPMC Content and Polyphenol Levels in Carbopol Containing EKG Gel Preparations

The RP-HPLC method was performed to analyze the EPMC contained in the EKG gel preparation. EPMC was identified if the gel preparation showed chromatogram peaks with retention times resembling standard EPMC (Tokyo Chemical Industry Co., Ltd., Tokyo, Japan) [10]. The HPLC chromatogram proved that 50 ppm of EPMC was eluted at 5.885 min, while F1 and F2 showed peaks at 6.041 and 6.146 min, respectively (Figure 2), thus strongly confirming the presence of EPMC in the F1 and F2 gel preparations.

Polyphenol levels in the F1 and F2 gels were also identified using an ultraviolet–visible spectrophotometer and the Folin–Ciocâlteu method. The polyphenol level in F1 was 1201.2557 mg/kg, and in F2, it was 1849.1504 mg/kg.

### 3.3. The Hedonic Tests on the Best EKG Gel Preparations

The hedonic tests showed that no respondents answered with “very unlikely” in all terms of both of the gel formulas (Figure 3). The answer option “likely” was chosen by most of the respondents. F1 is more favored by respondents compared to F2 in all parameters.

## 4. Discussion

The present study developed the optimal topical gel formula containing 5% and 10% of EKG for oral mucosal ulceration. The main findings of this study are that (1) Carbopol 934 is the best gelling agent for EKG gel preparations as proven by its stability during 14 days of storage; (2) RP-HPLC chromatograms indicated that EPMC was identified in Carbopol 934-based gels containing 5% and 10% EKG at retention time of 6.056 and 6.146 min, respectively, with polyphenol levels of 1201.2557 mg/kg and 1849.1506 mg/kg, respectively; and (3) the hedonic test, performed on 30 respondents, confirmed that 5% EKG gel was the most sensorially accepted by the respondents.

The 5% and 10% EKG in the Carbopol 934 base demonstrated good stability concerning physicochemical parameters, namely, taste, homogeneity, pH, viscosity, and spreadability, as well as hedonic parameters, namely, aroma, color, texture, and convenience, on day 14 compared to day 0. This stability is expected due to Carbopol 934’s properties, facilitating dispersion in the water despite its low concentration. To prevent complete dissolution in water, a neutralizing agent like TEA (triethanolamine) is essential. TEA neutralizes and stabilizes Carbopol 934, forming a gel mass. It also neutralizes the acidity of Carbopol 934, resulting in a thick and clear gel. A proper concentration of Carbopol 934 for the gel formula is in the range of 0.5–2%. As more Carbopol 934 is added, the viscosity increases, and vice versa. The increase in viscosity during storage can be caused by the expansion of the polymer in the gel preparation and increases the bond density between polymers [26,27]. Our study used 1% Carbopol 934 as a gelling agent, although there were significant differences when compared to several commercial preparations, which showed good stability in physical properties and still met the requirements [28].

The ideal gel base for pharmaceutical preparations is inert. Carbopol 934 and Na-CMC are commonly used as gelling agents in gel formulas due to their ability to provide good viscosity. Carbopol 934 polymer is a hydrophilic polymer with a poly-acrylic acid structure. Carbopol 934 is a strong gelling agent, so only a small concentration is needed to form a gel. Na-CMC is the gelling agent from cellulose derivatives and is often used because it produces a gel that is neutral and has a stable viscosity. However, the weakness of using cellulose derivatives is their susceptibility to enzymatic degradation by organisms, resulting in a decrease in viscosity. To prevent this, methylparaben is needed as a preservative agent [27]. In this study, the Na-CMC was not suitable as a gelling agent for EKG, because it exhibits poor physical stability.

The pH test is crucial for analyzing the acidity level of the gel preparation to ensure it does not irritate the oral mucosa [29]. The pH test results show that all of the tested gel formulas met the pH requirements for oral care gel preparations (pH range of 5.5–7.9). The pH values of F1 and F2 increased on the 14th day, while the pH value of the F3 gel preparation decreased. The pH value of the preparation depends on the constituent components of either the active substance or the additives used in the formulation. Changes in the pH value during storage were affected by high temperatures during manufacture or storage, which produce acids or bases. Changes in the pH value during storage indicate a reaction or damage to the constituent components in the preparation and decrease or increase the pH value of the preparation. Environmental factors, such as temperature and inadequate storage conditions, can also influence pH value alterations during storage. On the other hand, the combinations of extracts and mucoadhesive agents can decrease the physical stability, due to oxidation [28].

The spreadability test serves to determine the ability of the gel preparation to spread in the oral mucosa. The spreadability differences of the gel will affect the rate of diffusion of the active substance across the membrane. The spreadability of the gel is also affected by the viscosity. The viscosity of the gel is inversely proportional to the resulting spreadability [27]. The spreadability value of the F3 gel preparations increased during the 14-day shelf life. This is because the gel base cannot retain the water content after being reacted with the extract, then the gel preparation becomes aqueous over time [30,31]. The spreadability value is also influenced by the composition of the materials used, namely, gelling agents and humectants. A gelling agent will form a structural network which is an important factor in the gel system. Humectants will maintain the stability of the gel preparation by absorbing moisture from the environment and reducing water evaporation from the preparation [26].

EKG in Carbopol 934-based (F1 and F2) contained polyphenol levels of 1201.2557 GAE mg/kg (1.20 GAE mg/g) and 1849.1504 GAE mg/kg (1.85 GAE mg/g), respectively. Previous research reported that the total polyphenol content in the ethanolic extract of K. galanga rhizome originated from the highlands and lowlands of the Purbalingga area was 17.92 GAE mg/g and 24.85 GAE mg/g [32]. This difference is likely because we measured the total phenolic content only from the 5% and 10% of EKG contained in the gel formula. Variances in geographical conditions of plant growth, harvest time, and processing procedures for synthesizing the extract can lead to differences in secondary metabolite content [10]. Phenols are compounds containing a hydroxyl group attached directly to an aromatic hydrocarbon ring group. The antioxidant activity of phenolic compounds arises from their ability to form phenoxide ions which can give one electron to free radicals. The phenol compound group also has several benefits such as anti-inflammatory and antioxidant effects [33,34,35]. Thus, the polyphenols in the F1 and F2 gels are thought to reduce the inflammatory process and promote wound healing.

Figure 2 shows that the EPMC was identified in F1 and F2 (5% and 10% EKG in Carbopol 934-based gel preparations), with a retention time of 6.041 and 6.146 min, respectively, resembling the retention time of standard EPMC at 5.885 min. This is in line with our previous results, in which EPMC was detected on the EKG using the reverse-phase HPLC method [10]. Thus, it indicates that EPMC, as the major secondary metabolite contributing to anti-inflammatory activity, is still present in the gel preparation.

Our previous study results also proved the effects of kaempferol (KAE), EPMC, and EKG on prostaglandin production by inhibiting the COX-2 enzyme (in vitro study). The in-silico study also showed that KAE and EPMC can occupy catalytic sites of COX-2 by building hydrogen bonds with important amino acid residues. The interaction between EPMC and the Ser530 residue in the COX-2 catalytic site is similar to that of celecoxib, a selective COX-2 inhibitor. COX-2 inhibition can decrease prostaglandin synthesis, a mediator of inflammation [9]. Considering the presence of EPMC and polyphenol contents in the F1 and F2 gel formulas, it can be estimated that these gels are suited to being developed as anti-inflammatory drugs.

Figure 3 shows the histogram of the hedonic test, which is a test in organoleptic sensory analysis. The results of the hedonic test aid in determining the preferred formula for further development, considering consumer preferences [16,17]. Based on the overall hedonic test data, it can be concluded that the better gel formula is F1, Carbopol 934 contained 5% EKG. This is because the EKG dosage in F2 is higher than in F1. The higher doses of extract lead to a stronger aroma and a darker color in F2 [18]. Since not everyone favors herbal aromas, this could explain why F2 is less favored by respondents compared to F1, especially among the predominantly young respondents who are generally less inclined toward herbal scents.

The release of the drug in the oral mucosa is considered important due to the presence of saliva in the mouth; thus, an oral gel preparation requires vehicle material that overcomes this condition. The flow of saliva dissolves the active drug substance before it is absorbed into the inflamed tissue, thereby reducing the drug’s activity. An oral care gel formula should have mucoadhesive properties, which are also influenced by various factors, e.g., pH. Moreover, the physical stability of the gel preparation is an important requirement. The hydrogel formula containing active drug substances acts as a mucoadhesive, creates a protective layer on the oral mucosa surface, and retains the active drug substance to act longer, categorized as type 3, namely adhesion between artificial materials and biological tissue [36].

## 5. Conclusions

Carbopol 934 is the best mucoadhesive polymer for EKG gel formulation. Carbopol 934-based EKG positively contains EPMC and polyphenols. Moreover, by considering the acceptance and satisfaction of the respondents, Carbopol 934-based 5% EKG could potentially be further developed as a topical anti-oral mucosal ulcer drug for clinical purposes.

## Figures and Tables

**Figure 1 pharmaceutics-16-00421-f001:**
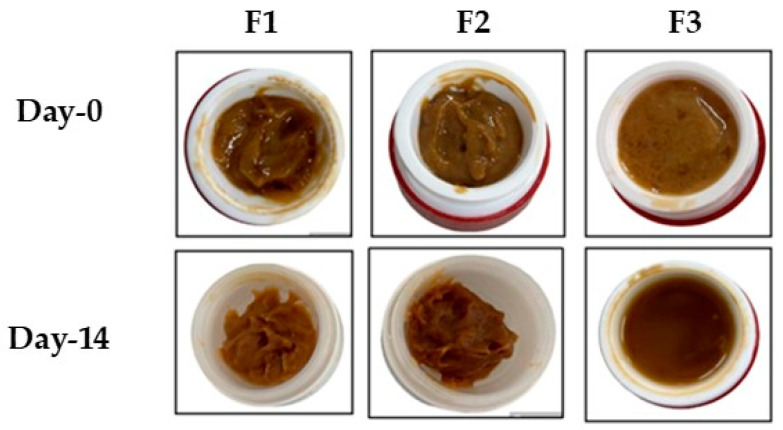
Carbopol 934 (F1 and F2) and Na-CMC (F3) containing EKG in gel formulation, observed on day 0 and day 14.

**Figure 2 pharmaceutics-16-00421-f002:**
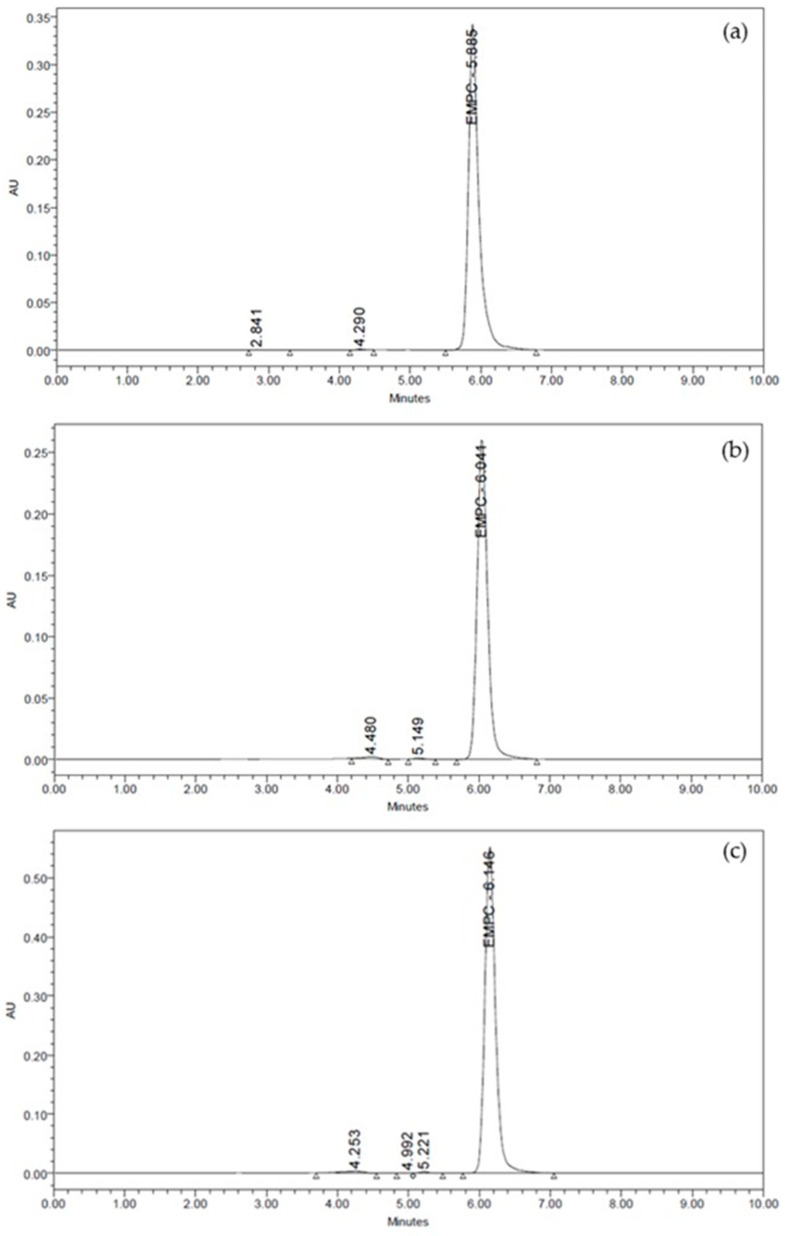
Comparison of RP-HPLC chromatograms of standard EPMC (**a**), F1 (**b**), and F2 (**c**).

**Figure 3 pharmaceutics-16-00421-f003:**
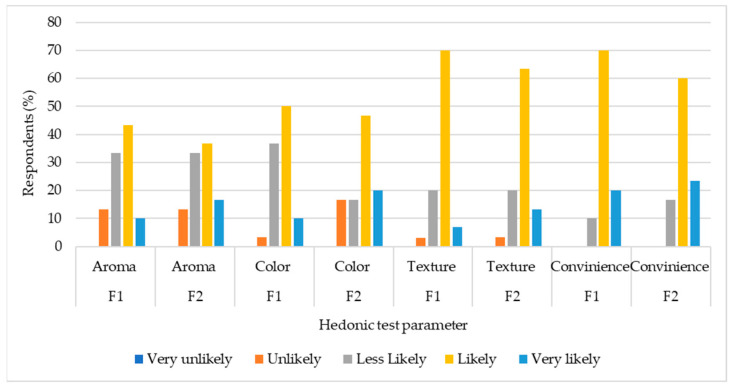
Histogram of the hedonic test for F1 and F2.

**Table 1 pharmaceutics-16-00421-t001:** Organoleptic test on EKG formulas.

Group	Organoleptic Test		Interpretation
	Day 0	Day 14	
F1	Thick texture, dark-brown color, with a pungent odor of the rhizome, bitter and spicy taste	Thick texture, dark-brown color, with a pungent odor of the rhizome, bitter and spicy taste	Stable
F2	Thick texture, dark-brown color, with a pungent odor of the rhizome, bitter and spicy taste	Thick texture, dark-brown color, with a pungent odor of the rhizome, bitter and spicy taste	Stable
F3	Thick texture, dark-brown color, with a pungent odor of the rhizome, bitter and spicy taste	The semisolid and the liquid phases were separated, dark-brown color, with a pungent odor of the rhizome, bitter and spicy taste	Unstable

**Table 2 pharmaceutics-16-00421-t002:** Homogeneity test on EKG gel formulations.

Group	Homogeneity		Interpretation
	Day 0	Day 14	
F1	Good Homogeneity	Good Homogeneity	Stable
F2	Good Homogeneity	Good Homogeneity	Stable
F3	Good Homogeneity	Bad Homogeneity	Unstable

**Table 3 pharmaceutics-16-00421-t003:** pH test on EKG gel preparations.

Group	pH Test		*p*-Value	Interpretation
	Day 0	Day 14		
F1	6.19	7.94	0.528	Not significantly different
F2	6.35	6.98		
F3	5.65	5.44		

Notes: The test was performed using paired *t*-test (*p* < 0.05).

**Table 4 pharmaceutics-16-00421-t004:** Viscosity test on EKG gel preparations.

Group	Viscosity (mPas)		*p*-Value	Interpretation
	Day 0	Day 14		
F1	351	425	0.533	Not significantly different
F2	377	420		
F3	410	154		

Notes: The test was performed using paired *t*-test (*p* < 0.05).

**Table 5 pharmaceutics-16-00421-t005:** Spreadability test on EKG gel preparations.

Group	Spreadability (cm)		*p*-Value	Interpretation
	Day 0	Day 14		
F1	4.2 cm	4.2 cm	0.191	Not significantly different
F2	4 cm	4 cm		
F3	3.95 cm	5.5 cm		

Notes: Statistical analysis was performed using paired *t*-test (*p* < 0.05).

**Table 6 pharmaceutics-16-00421-t006:** pH test on F1 and F2 compared with commercial oral care gel preparations.

Group	pH Value	*p*-Value	Interpretation
F1	7.820 ± 0.012	0.000	Significantly different
Gengigel gel^®^	6.183 ± 0.015		
Aloclair gel Plus^®^	6.397 ± 0.015		
Kincare gel preparations^®^	5.900 ± 0.100		
F2	6.977 ± 0.035	0.000	Significantly different
Gengigel gel^®^	6.183 ± 0.015		
Aloclair gel Plus^®^	6.397 ± 0.015		
Kincare gel preparations^®^	5.900 ± 0.100		

Notes: Data are presented as mean ± SD. Statistical analysis was performed using one-way ANOVA (*p* < 0.05).

**Table 7 pharmaceutics-16-00421-t007:** Viscosity test on F1 and F2 compared with commercial oral care gel preparations.

Group	Viscosity (mPas)	*p*-Value	Interpretation
F1	422.667 ± 2.516	0.000	Significantly different
Gengigel gel^®^	622.000 ± 2.000		
Aloclair gel Plus^®^	700.000 ± 10.000		
Kincare gel preparations^®^	722.667 ± 2.516		
F2	419.000 ± 1.000	0.000	Significantly different
Gengigel gel^®^	622.000 ± 2.000		
Aloclair gel Plus^®^	700.000 ± 10.000		
Kincare gel preparations^®^	722.667 ± 2.516		

Notes: Data are presented as mean ± SD. Statistical analysis was performed using one-way ANOVA (*p* < 0.05).

**Table 8 pharmaceutics-16-00421-t008:** Spreadability test on F1 and F2 compared with commercial oral care gel preparations.

Group	Spreadability (cm)	*p*-Value	Interpretation
F1	4.21 ± 0.01	0.015	Significantly different
Gengigel gel^®^	4.05 ± 0.05		
Aloclair gel Plus^®^	3.86 ± 0.03		
Kincare gel preparations^®^	3.72 ± 0.03		
F2	4.00 ± 0.10	0.026	Significantly different
Gengigel gel^®^	4.05 ± 0.05		
Aloclair gel Plus^®^	3.86 ± 0.03		
Kincare gel^®^	3.72 ± 0.03		

Notes: Data are presented as mean ± SD. Statistical analysis was performed using Kruskal–Wallis test (*p* < 0.05).

## Data Availability

Data can be requested from the authors if needed via email correspondence.

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
