# Peer review of "Formulation and Evaluation of Mucoadhesive Oral Care Gel Containing Kaempferia galanga Extract"

_pharmaceutics, 2024, doi:10.3390/pharmaceutics16030421_

Round 1

Reviewer 1 Report

Comments and Suggestions for Authors

Journal: MDPI Pharmaceutics, Type: Research

Title: Formulation and Evaluation of Mucoadhesive Oral Care Gel Containing Kaempferia galanga Extract

Manuscript ID pharmaceutics-2902388

Comments

This research work aims to development of oral gel for oral care containing prepared Kaempferia galanga Extract. The work has serious implications in terms of formulation, composition and characterization. It lacks novelty and even lacks scientific representation. Only three batches F1 to F3 formulated and brief physico chemical characterization is done.

The constructive comments are as follows.

1.       Clarity of Rationale/Problem statement must be there in abstract as well as at the introduction,

The current explanation: Taking everything into consideration, our research aims to obtain the optimal gel formula of EKG, focusing on both physical properties and stability, as well as evaluating

the preference of the gel by the panelists. The polyphenol content in EKG gel was also determined to ensure the presence of these active metabolites, is not enough to provide clarity on the novelty of this work.

2.       The title is Formulation and Evaluation of Mucoadhesive Oral Care Gel, and the author talks about physical stability as a problem. Is the release of drugs important? Is there any target for it? What about mucoadhesion?

3.       Line 72-78: Recently several non-steroidal anti-inflammatory topical drugs have become commercially available in Indonesia, indicated for oral mucosal inflammation, among them is Gengigel® containing 0.2% hyaluronic acid, Kincare® containing 0.24% hyaluronic acid, and aloe vera extract (Aloe vera), as well as Alolair® Plus which also contains Aloe vera extract and sodium hyaluronate. These three commercial products possess mucoadhesive properties that can increase the formation of a protective layer over the oral mucosal ulcer, moisturize, and enhance wound healing. Please cite this paragraph. Is formulated product compared with any one of these?

4.       Why carbopol selected in place of natural polymer? There are good natural mucoadhesive polymers available.

5.       Author use carbopol 934 or 934P grade?

6.       No need for lab detail: Table 1 shows the gel formulas that have been tested, consisting of 3 formulations: F1, F2, and F3. The preparation of EEKG gel preparations based on Na- CMC gel was carried out at the Cosmetic Laboratory, Faculty of Pharmacy, Padjadjaran University in February - March 2023. Additionally, table 1 shows 2 batches with carbopol and one with Na CMC. Please be clear about what was used in formulations.

7.       Why there is a range in Table 1? No need a specified amount used.

Comments on the Quality of English Language

few sentences are not clear. 

Reviewer 2 Report

Comments and Suggestions for Authors

The present pharmaceutics-2902388 manuscript, by Indah Suasani Wahyuni et al, entitled "Formulation and Evaluation of Mucoadhesive Oral Care Gel Containing Kaempferia galanga Extract" refers to the authors' work on obtaining the optimal gel formula of K. galanga 55 L. (EKG), focusing on both physical properties and stability, as well as evaluating the preference of the gel by the panelists.

After considerable experimentation the authors found that Carbopol 934 is the best mucoadhesive polymer for EKG gel formulation. Carbopol 934-based EKG positively contains EPMC and polyphenols. Moreover, by considering the acceptance and satisfaction of the respondents, Carbopol 934-based containing 5% EKG is prospective to be further developed as a topical anti-oral mucosal ulcer drug for clinical purposes.

All data were analyzed using paired t-tests, one-way ANOVA, and Kruskal-Wallis. 

The article is concisely written, well documented and of interest to both the cognizant and non-cognizant reader. A point that could be emphasized more, in the Introduction of the revised version of the manuscript, is the fact that the intraperitoneal administration of KGE is known to decrease the number of tumor cells and extends the lifespan of mice inoculated with Ehrlich ascites tumor cells (EATCs). KGE decreases the survival rate and fragmentation of the nucleus in EATCs (e.g. vide doi: 10.1016/j.heliyon.2023.e17588)

This is an important information that has to be included in the manuscript, as natural compounds can be developed into novel anti-cancer drugs, devoid of the serious side effects the currently used chemotherapeutics are linked with.

Reviewer 3 Report

Comments and Suggestions for Authors

The presented manuscript is devoted to the creation of an oral mucoadhesive gel containing an extract of Kaempferia galanga.

The design of such formulations with improved biopharmaceutical properties represents an important challenge in modern biomedical and colloid chemistry.

The work is of an applied field, the authors have optimized the formulation of the gel, provided test results for stability, viscosity, pH, etc. Based on the data obtained, the authors choose the optimal formulation.

At the moment, the work does not sufficiently reflect scientific novelty. What is the fundamentally new result, what new patterns were discovered by the authors of the manuscript? I definitely recommend to enrich the Discussion section.

Authors should provide more careful description of experiments involving volunteers and provide ethics committee approval or justification for non-applicability.

Round 2

Reviewer 1 Report

Comments and Suggestions for Authors

There are a few points still missing, like mucoadhesion study data and strong rational statement. However, the work is significantly modified and efforts are there to improve the quality. 

I am satisfied with the revision. 

Reviewer 3 Report

Comments and Suggestions for Authors

The authors have considered all remarks and improved the manuscript

The manuscript can be accepted in the present form